

# Phytoplankton communities determine the spatio-temporal heterogeneity of alkaline phosphatase activity: evidence from a tributary of the Three Gorges Reservoir

**Yijun Yuan[1,2], Yonghong Bi[1,*] and Zhengyu Hu[1]**

[1] Key Laboratory of Algal Biology, Institute of Hydrobiology, Chinese Academy of Sciences, 430072, Wuhan, China

[2] University of Chinese Academy of Sciences, 100049, Beijing, China

*Corresponding author:    E-mail address: biyh@ihb.ac.cn.

Tel/fax: +86 27 68780016.

**Abstract.** In order to know the role of phytoplankton communities in the distribution pattern of alkaline phosphatase activity (APA), monthly investigation was conducted in the Xiaojiang River, a tributary of the Three Gorges Reservoir (TGR). Different APA fractions ($APA_T$, $APA_{<0.45\mu m}$, $APA_{0.45-3\mu m}$ and $APA_{>3.0\mu m}$), environmental parameters, and phytoplankton communities were screened synchronously. Significant spatio-temporal differences of APA with the highest value in summer and the lowest in winter ($P<0.05$) were observed. The annual average $APA_T$ ranged from 7.78-14.03 $nmol \cdot L^{-1} \cdot min^{-1}$ with the highest in the midstream and the lowest in the estuary. The dominant phytoplankton species in summer and winter were Cyanophyta and Bacillariophyta, respectively. The mean cell density in the midstream and in the estuary were $5.2 \times 10^7 cell \cdot L^{-1}$ and $1.4 \times 10^7 cell \cdot L^{-1}$, respectively. That $APA_{>3.0\mu m}$ were significantly higher than $APA_{0.45-3\mu m}$ indicated phytoplankton was the main contributor to alkaline phosphatase. Correlation analysis indicated the dominant species and cell density could determine the distribution pattern of APA. Turbidity (Turb), total phosphorus (TP), chemical oxygen demand (COD), water temperature (WT), pH and chlorophyll *a* (Chl *a*) were proved to be positively correlated with APA; soluble reactive phosphorus (SRP), conductivity (Cond), transparency (SD) and water level (WL) were negatively correlated with APA. It was concluded that spatio-temporal heterogeneity of APA determined by phytoplankton communities was related to water temperature and hydrodynamics.

## 1 Introduction

Alkaline phosphatase (APase) can hydrolyze broad spectrum phosphomonoesters (Kuenzler and Perras, 1965; Tanaka *et al*., 2008) and associate with cells surfaces of microbial organisms (Gonzalez *et al*.,1998). Both phytoplankton and bacteria can secrete extracellular APase which enables them to use organic P esters as a source of P for





compensation of P deficiency (Ivancic *et al*., 2009). The significant seasonal and regional
variations of APA were found (Zhang *et al*.,2013). The inverse proportion of alkaline
phosphatase activity (APA) to SRP concentration was summarized as "induction-repression"
mechanism (Jansson *et al*., 1988). APase plays an important role in the aquatic phosphorus
cycling.
Relationship between APA and phytoplankton has been paid more attention since 1960s
(Perry, 1972; Kuenzler, 1965). Kalinowska tried to figure out the major contributor of APase
through membrane filtration method (Kalinowska, 1997). Even if size fractionation by
filtration is never completely absolute (i.e., overlapping size), it still provides useful insights
on the major microorganisms possibly contributing to APA. Because of the higher biomass of
phytoplankton than bacteria in the open ocean and coastal areas, the phytoplankton makes a
bigger contribution to the hydrolysis of DOP to DIP (Nausch, 1998). Therefore,
phytoplankton contributed greatly to APA production and was significantly influenced by P
bioavailability. Production of extracellular phosphatases has been detected in many
phytoplankton species (Rengefors *et al*., 2001; Cao *et al*., 2005; Strojsova *et al*., 2008).
Various taxa are exhibiting differences in the presence, localization and labelling pattern of
phosphatases. Both seasonal and short-term variations also have been detected in enzyme
activity of phytoplankton (Strojsova and Vrba, 2009). Enzyme-labeled fluorescence (ELF)
analysis revealed pronounced differences in the makeup of phytoplankton responsible for
APA in San Francisco and Monterey bays (Nicholson *et al*., 2006). Though many studies
have been conducted to screen APase in different water bodies, little information could be
obtained in the Three Gorges Reservoir (TGR).
TGR is the biggest deep river-type reservoir in the world. More than 170 tributaries
carrying runoff and bringing nutrients and pollutants into it, which affected the trophic status
and resulted in phytoplankton blooms in many bays of the TGR. To date, little information of
APA in the TGR and its tributaries could be found. Due to the complicated relationship
between APA and ecological factors, it is necessary to screen the distribution pattern of APA
in the TGR. Xiaojiang River is one of the tributaries in the TGR, which was suffered from
phytoplankton blooms frequently like other tributaries; eutrophication in Xiaojiang River is
very serious after the Three Gorges Dam (TGD)'s impoundment since 2003 (Li *et al*, 2009).
In this study, Xiaojiang River was selected as the delegate of the tributary in the TGR,
phytoplankton and APA in Xiaojiang River were screened. It was assumed that the
phytoplankton community successions may lead to the spatio-temporal heterogeneity of
alkaline phosphatase activity. In order to verify this hypothesis, monthly investigation was
conducted, different APA fractions ($APA_T$, $APA_{<0.45\mu m}$, $APA_{0.45\text{-}3\mu m}$ and $APA_{>3.0\mu m}$),
environmental parameters and phytoplankton communities were screened synchronously. The
role of phytoplankton communities in the spatio-temporal heterogeneity of APA and its





influence factors in the Three Gorges Reservoir were demonstrated. The results of this study
can help to know how APA production changes with phytoplankton communities'
successions in TGR.

## 2 Materials and methods

### 2.1 Samples and sites

Xiaojiang River, a tributary of the TGR, originates from Kaixian, Chongqing Municipality
with a length of 180 km and watershed area of 5172.5 km$^2$. It flows from north to south;
entering into the TGR in Yunyang County. The distance from the estuary to the TGD is 248
km.
Surface water samples (0.5m) were collected with a Van Dorn sampler at seven sampling
sites (XJ, HS02, HS01, GY02, GY01, QM02, QM01) (Fig.1) monthly from October 2013 to
September 2014. Water temperature (WT), pH, dissolved oxygen (DO) and conductivity
(Cond.) were measured using a YSI model Professional Plus multiparameter probe (USA);
Transparency (SD) was measured with a Secchi disk; and turbidity (Turb.) was measured
with a WGZ-B turbidmeter (XinRui, Shanghai). Water level (WL) was recorded by GPS *in*
*situ*. Concentrations of chlorophyll *a* (Chl *a*), total phosphorus (TP), soluble reactive
phosphorus (SRP), chemical oxygen demand (COD) were analyzed in 24 h. Samples for
quantitative phytoplankton analyses were fixed with neutral Lugol's solution, and
concentrated after 48 h sedimentation (Utermohl, 1931).

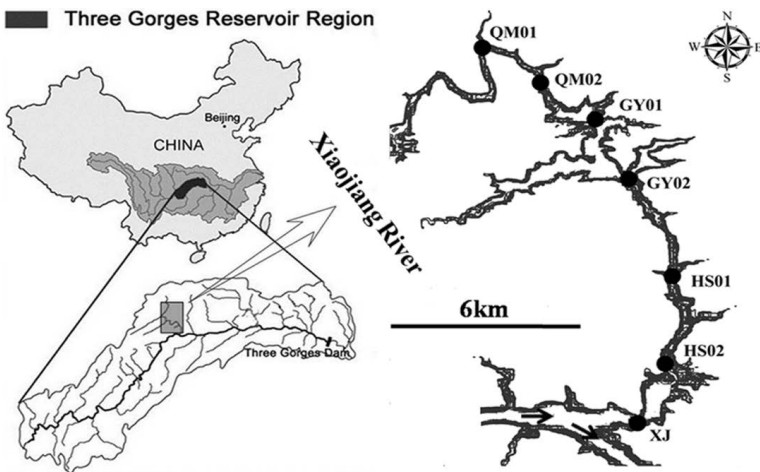


**Figure1.** Maps of the location of the Three Gorges Reservoir Region, and the sampling sites in
the Xiaojiang River

### 2.2 Measurement of APA

APA was measured using a modified procedure (Gage and Gorham, 1985; Boon, 1989). A
total of 2ml water samples were incubated at 37 ℃ for 4h in the pr
Tris-HCl buffer (pH=8.5) and 2ml 0.3mM p-nitronphenylphosphate (p-NPP) as substrate,





subsequently, 0.1ml 0.1M NaOH was added into the mixture after 4h. The release of
p-nitrophenol from p-nitronphenylphosphate was determined by absorbance at 410nm using a
spectrophotometer (TU-1810), and APA was calculated in $nM \cdot L^{-1} \cdot min^{-1}$. APA was
determined in unfiltered water ($APA_T$) and water samples filtered through 0.45 (dissolved
alkaline phosphatase activity, $APA_{<0.45\mu m}$) and 3.0μm membrane filters ($APA_{<3.0\mu m}$). The
activity in algal fraction ($APA_{>3.0\mu m}$) and in bacterial fraction ($APA_{0.45-3.0\mu m}$) were calculated as
follows: $APA_{>3.0\mu m}= APA_T － APA_{<3.0\mu m}$, $APA_{0.45-3.0\mu m}= APA_{<3.0\mu m} － APA_{<0.45\mu m}$ (Chrost *et al*.,

1984).

**2.3 Measurement of SRP, Chl *a* , TP, COD and phytoplankton quantification**
Water samples used for the Chl *a* measurement were filtered with Whatman GF/C filter,
then the residuals on the filter were extracted using 90% acetone solution in the darkroom for
24 h at 4°C, and Chl *a* was analyzed spectrophotometrically (A.P.H.A, 1995). The
concentrations of SRP were measured after all water samples were filtered through
pre-washed filters (Whatman GF/C, glass microfiber filters).The concentrations of SRP, total
phosphorus (TP) and chemical oxygen demand (COD) were analyzed according to the
standard methods (A.P.H.A, 1995). Phytoplankton was quantified at 400× magnification with
a light microscope (OLYMPUS BX41). The identification of phytoplankton species is
according to Hu and Wei (Hu and Wei, 2006).
**2.4 Statistical analysis**
Statistical analysis was carried out using the SPSS 13.0 package. Variance analysis
(one-way ANOVA) was used to compare the means of APA in different seasons and
sampling sites. Non-parametric correlation (Spearman) analyses were employed for
determining relationships among $APA_{<0.45\mu m}$, $APA_{0.45-3\mu m}$, $APA_{>3.0\mu m}$, $APA_T$ and the
environmental factors. Detrended correspondence analysis (DCA) of the size-fractioned
APA and environmental data was performed using CANOCO version 4.5 to determine
whether linear or unimodal ordination methods should be applied. Before the analysis, the
abiotic and biological data were transformed by log(x+1). Redundancy analysis (RDA) was
performed to get an approximate ordering of the size-fractioned APA's optima for
environmental variables. The significance of canonical axes and environmental variables to
explain the variance of the size-fractioned APA was tested using Monte Carlo simulations
with 499 permutations.
**3. Results**
**3.1 $APA_T$ distribution pattern**
The $APA_T$ ranged from 1.19-47.6 $nmol \cdot L^{-1} \cdot min^{-1}$(Fig.2). The lowest level of $APA_T$ was
observed in winter. Besides, the average $APA_T$ in summer and autumn were significantly
higher than in other seasons ($P<0.05$). Meanwhile, significant difference between summer




and autumn were not detected ($P>0.05$). The mean water level was high in winter($169.7\pm4.5$
m) and low in summer($149.3\pm3.1$ m), the variations of water level presented different trends
with that of $APA_T$ at temporal scales.

The highest value of annual average $APA_T$ in GY01 and lowest in XJ were also showed in

Fig.2. No difference of $APA_T$ among the seven sites was observed in winter and spring
($P>0.05$). The average $APA_T$ of GY01 in summer and autumn are significantly higher than
those of HS02 and XJ ($P<0.05$).

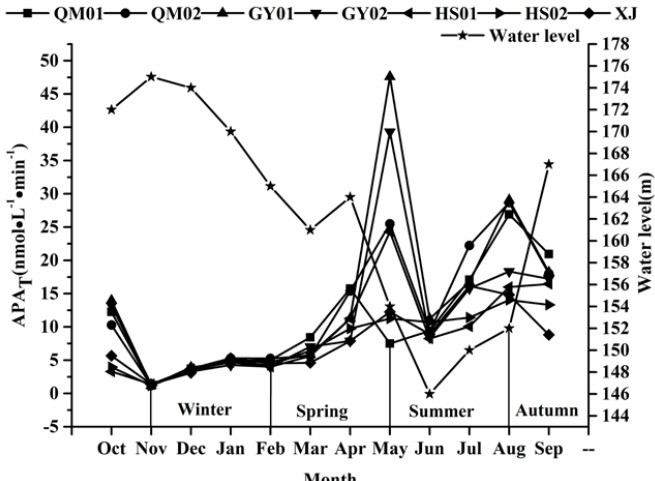


**Figure 2.** Seasonal variations in $APA_T$ concentrations in different sample sites and water level

of the Xiaojiang River

**3.2 Size-fractionation of APA**

The average size-fractionated APA indicated that $APA_{<0.45\mu m}$ accounted for the major

portion of $APA_T$, whereas the average $APA_{>3.0\mu m}$ are significantly higher than $APA_{0.45-3\mu m}$
($P<0.05$) (Fig.3a). The average $APA_{>3.0\mu m}$ accounted for 28.1% of $APA_T$ and $APA_{0.45-3\mu m}$
accounted for 16.7%. In addition, the size-fractionated APA ($APA_{<0.45\mu m}$, $APA_{0.45-3\mu m}$ and
$APA_{>3.0\mu m}$) in summer and autumn are significantly higher than those in winter ($P<0.05$).

At spatial scales, the average $APA_T$ consisted of 30.2% $APA_{>3.0\mu m}$ and 20.4% $APA_{0.45-3\mu m}$ in

all sites. The $APA_{<0.45\mu m}$ kept a relatively stable and high level. Both $APA_{0.45-3\mu m}$ and
$APA_{>3.0\mu m}$ in midstream (GY01) are higher than those in estuary (XJ) ($P<0.05$).



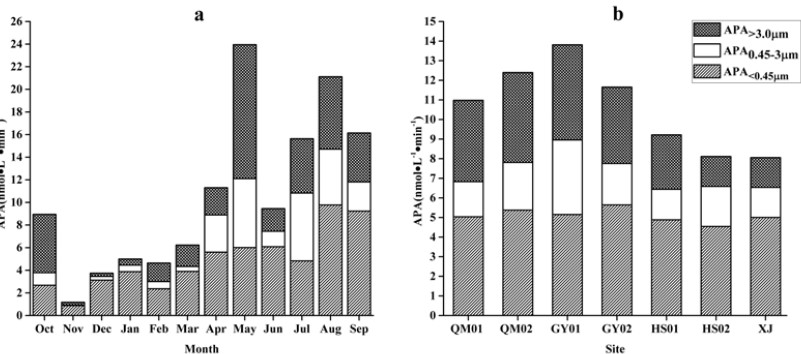


**Figure 3.** Seasonal (a) and spatial (b) variations of average size-fractionated APA in the Xiaojiang
River. $APA_{>3.0\mu m}$: the alkaline phosphatase activity in algal fraction; $APA_{0.45-3.0\mu m}$: the alkaline
phosphatase activity in bacterial fraction; $APA_{<0.45\mu m}$ : dissolved alkaline phosphatase activity
**3.3 Phytoplankton communities**
Bacillariophyta was the dominant group in winter and spring (72.7% in average, Fig.4). In
summer and autumn, phytoplankton mainly consisted of Cyanophyta (65.6% in average)
except the Cryptophyta accounted for 88.4% in August. The mean algal cell density was the
highest in July 2014 ($1.27 \times 10^8$ cell · $L^{-1}$), and the lowest in January 2014 ($1.3 \times 10^6$ cell · $L^{-1}$).
The cell density was higher in summer and autumn than in spring and winter.

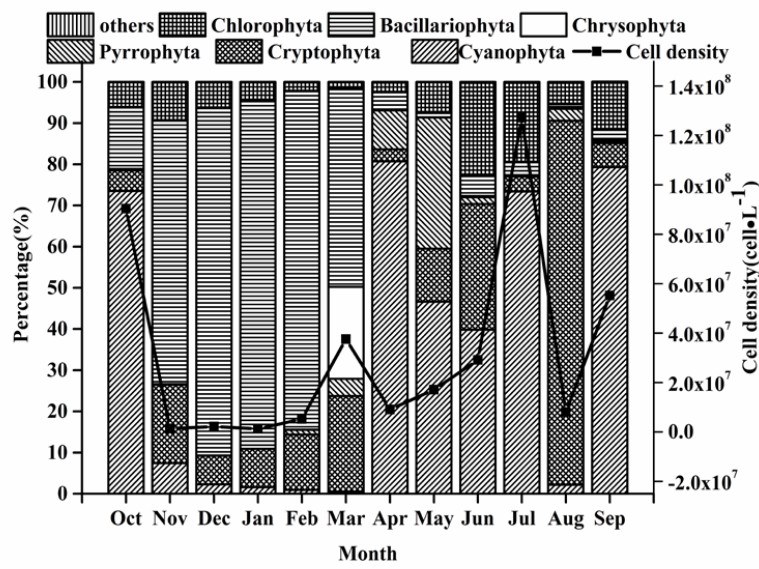


**Figure 4.** Seasonal variations of algal composition and algal cell density in the Xiaojiang River
**3.4 Spatio-temporal characteristics of Chl $a$ and environmental parameters**
Significant seasonal variations of Chl $a$ and environmental parameters could be observed



(Fig.5). The values of Chl *a*, TP, COD in spring were apparently higher than the values of
other seasons, because the river suffered a *Microcystis* sp. bloom in May, which also resulted
in the minimum values of SRP and SD emerged. The levels of TP, COD, Chl *a*, WT, Turb,
DO and pH stayed low in winter, contrary to the values of SRP and SD. The values of SRP
fluctuated more frequently than other parameters in different seasons. The concentrations of
SRP in estuary (XJ) were higher than in upstream. Chl *a* in estuary were higher than that in
upstream in May and the values was higher in upstream in March.











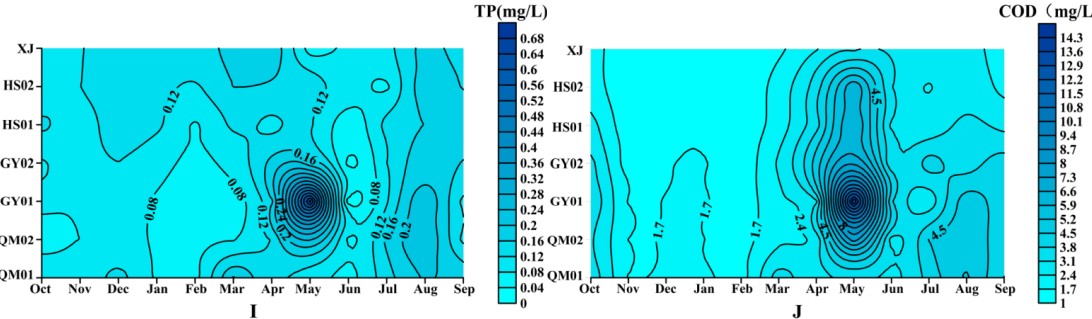

**Figure 5.** Temporal and spatial variations of A: chlorophyll *a* (Chl *a*) and other environmental parameters. B: soluble reactive phosphorus (SRP); C: water temperature (WT); D: transparency (SD); E: dissolved oxygen (DO); F: conductivity (Cond); G: pH; H: turbidity (Turb); I: total phosphorus (TP) and J: chemical oxygen demand (COD)

**3.5 Relationships between APA and environmental parameters**

SRP concentrations showed negative correlation to $APA_{<0.45\mu m}$ (Fig.6a), $APA_{0.45-3\mu m}$(Fig.6b), $APA_{>3.0\mu m}$(Fig.6c) and $APA_T$(Fig.6d). The Spearman correlations among environmental variables and $APA_{<0.45\mu m}$, $APA_{0.45-3\mu m}$, $APA_{>3.0\mu m}$ and $APA_T$ were presented in Table 1. Turb, TP, COD, WT, pH and Chl *a* were positively correlated with APA fractions. Cond., SD and WL were negatively correlated with APA.

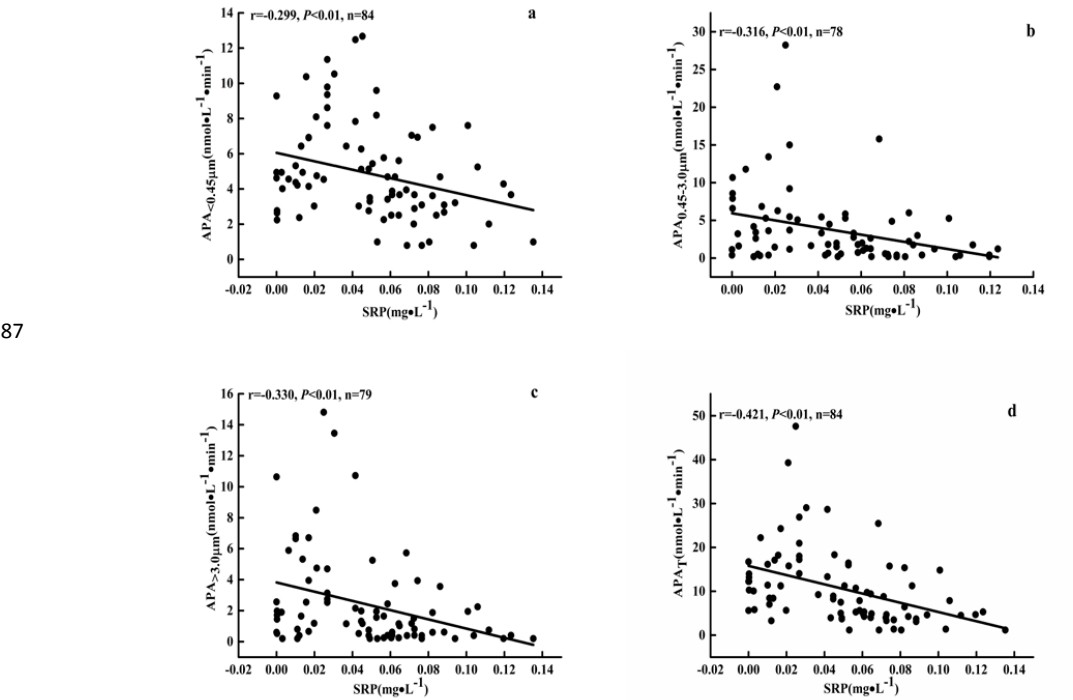

**Figure 6.** Relationship between soluble reactive phosphorus (SRP) concentrations and $APA_{<0.45\mu m}$ (a), $APA_{0.45-3\mu m}$ (b), $APA_{>3.0\mu m}$(c) and $APA_T$(d) in the Xiaojiang River





**Table 1.** Spearman correlations between APA  and 10 environmental variables: water
temperature (WT); chlorophyll a (Chl a); transparency (SD); dissolved oxygen (DO); conductivity
(Cond); pH, turbidity (Turb); total phosphorus (TP); chemical oxygen demand (COD); water level
(WL) in the Xiaojiang River

|  | $APA_T$(n=84) | $APA_{<0.45\mu m}$(n=84) | $APA_{>3.0\mu m}$(n=78) | $APA_{0.45-3\mu m}$(n=79) |
|---|---|---|---|---|
| WT | 0.642** | 0.562** | 0.404** | 0.609** |
| Chl *a* | 0.749** | 0.469** | 0.564** | 0.637** |
| SD | -0.844** | -0.815** | -0.586** | -0.698** |
| DO | 0.478** |  | 0.382** | 0.368** |
| Cond | -0.251* |  | -0.256* |  |
| pH | 0.405** |  | 0.271* | 0.271* |
| Turb | 0.858** | 0.834** | 0.582** | 0.753** |
| TP | 0.388** | 0.357** | 0.346** | 0.413** |
| COD | 0.858** | 0.684** | 0.646** | 0.751** |
| WL | -0.678* | -0.699* |  | -0.713** |

*$P<0.05$
**$P<0.01$
Redundancy analysis (RDA) was performed to analyze the relationship between
environmental parameters and size-fractionated APA. The ordination diagrams of
environmental variables and size-fractionated APA for axis 1 and axis 2 were shown in Fig.7.
The Monte Carlo test revealed that the first canonical axis and all canonical axes were
significant (F=25.932, P=0.002; F =3.086, P=0.002; 499 random permutation). For
environmental variables and size-fractionated APA, all canonical axes cumulatively explained
83.3% of the variance in APA–environment relationships, and the first two canonical axes
accounted for 26.5% and 31.5% of the variance separately. The first axis was positively
correlated with Chl *a* (0.65), DO (0.57), COD (0.65) and negatively correlated with SRP
(−0.41), SD (−0.38) and WL (−0.30). The second axis was mainly negatively correlated with
Cond (−0.13) and WT (−0.16). $APA_{<0.45\mu m}$ and $APA_{>3.0\mu m}$ was the major portion of $APA_T$.
$APA_T$, $APA_{>3.0\mu m}$ and $APA_{0.45-3\mu m}$ were located on the right-hand side of the biplot. They were
correlated negatively with WL, SD, SRP and Cond, and positively with other parameters.




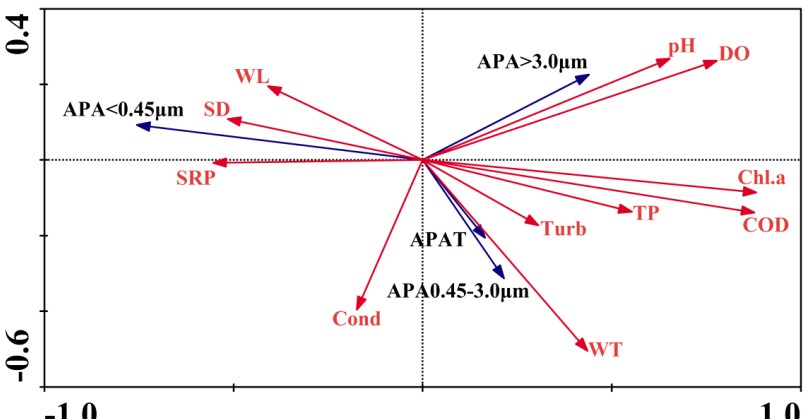

**Figure 7.** Biplot diagrams for RDA of the relationship between 11 environmental variables (red lines) and $APA_T$, $APA_{<0.45\mu m}$, $APA_{>3.0\mu m}$, $APA_{0.45\text{-}3\mu m}$ (blue lines.)

### 3.6 Relationships between $APA_{>3.0\mu m}$ and algal cell density

$APA_{>3.0\mu m}$ reached the highest in midstream (GY01) in May (28.24 nmol $\cdot$ $L^{-1}$ $\cdot$ $min^{-1}$), and undetectable in estuary (XJ) in December. Values ranged from 0.19-22.71 nmol $\cdot$ $L^{-1}$ $\cdot$ $min^{-1}$ at the other sites. The mean cell density was the highest in midstream (GY02, $5.2 \times 10^7$ cell $\cdot$ $L^{-1}$) and the lowest in estuary (XJ, $1.4 \times 10^7$ cell $\cdot$ $L^{-1}$). A significant positive relationship was found between $APA_{>3.0\mu m}$ and cell density among all sites (Fig.8).



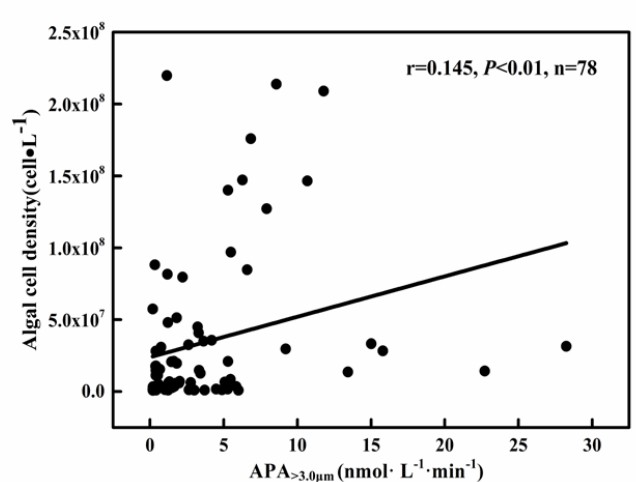

**Figure 8.** Relationships between $APA_{>3.0\mu m}$ and cell density in all sites

**4. Discussion**
APase has different sources, different kinds of bacteria, phytoplankton and zooplankton
can excrete extracellular phosphatase (Davey *et al*., 2001). Specific APA was related to
different phosphatase producing organisms. Phytoplankton associated phosphatase activity is
considered as a phosphorus deficiency indicator (Rose and Axler, 1997). The coarser fraction
($APA_{>3.0\mu m}$), mainly from algae, was conventionally defined as "algal APA" (Liu *et al*., 2012).
It was confirmed $APA_{>3.0\mu m}$ accounted for the major portion of total APA (55–87.9%) than
$APA_{0.45-3\mu m}$ (Cao *et al*, 2010). It could be deduced that the phytoplankton was the major
contributor of bulk APA based on the larger proportion of $APA_{>3.0\mu m}$(52.73%) than
$APA_{0.45-3\mu m}$(21.09%) (Wang *et al*., 2015). In this investigation, $APA_{>3.0\mu m}$ contributed in
average 28.1% in the $APA_T$, while bacterial APA accounted for 16.7%, APA in algal fraction
($APA_{>3.0\mu m}$) was also higher than that in bacterial fraction ($APA_{0.45-3\mu m}$). Therefore, the
phytoplankton contributed greatly to APA production that was consistent with the
observations in Wangyu River in China (Wang *et al*., 2015). Meanwhile, the dissolved APase
($APA_{<0.45\mu m}$) kept a relative stable and high level (53.4% of the $APA_T$). Some studies showed
that the dissolved APase represents a significant part of the total activity. For example, Labry
*et al* reported that dissolved APA represented 13% to 44% of $APA_T$ in the Bay of Biscay (on
the French Atlantic coast) (Labry *et al*., 2005). Higher proportions were recorded in the
northern Red Sea (42–74%) (Li *et al*., 1998). The dissolved APase can be liberated into the
environment through the lysis of dead phytoplankton cells and from cells damaged by
zooplankton grazing (Chrost, 1991). The high values may result from physical damage of
cells by water current and zooplankton grazing on phytoplankton. Nevertheless, some study



found that the dissolved APA might origin from bacteria (Hoppe and Ullrich, 1999). In order
to elucidate the origins of dissolved APA, Song *et al* microencapsulated the dissolved alkaline
phosphatase into reverse micellar media. Finally, they proved that the different behaviors of
dissolved phosphatase of surface and overlying water might be due to the different origins,
with the former being algae and the latter being bacterial (Song *et al*.,2005). It was deduced
that phytoplankton acted as the main contributor of dissolved APA in our research. Besides,
the positive relationships between APA and the environmental parameters that have been
treated as the indexes of the productivity and trophic status, such as Chl *a*, Turb and COD,
and the negative relationship between APA and SD can also indicate that the phytoplankton is
the main contributor of APA.
The introduction of Enzyme-labeled fluorescence (ELF) method can not only demonstrates
the existence of extracellular APase, but also localize where they are (Rengefors *et al*., 2001).
Different algal species showed significant different secreting ability of APase. Pyrrophyta,
Bacillariophyta, and Chlorophyta can easily produce extracellular phosphatase as evidenced
by ELFA labeling (Cao *et al*., 2010). In this study, phytoplankton communities were
dominated by Bacillariophyta in winter. The low $APA_{>3.0\mu m}$ during this period may result
from the low algal cell density of phytoplankton. Results in some shallow eutrophic lakes
revealed that the species belonging to Pyrrophyta were regularly phosphatase-positive, while
Bacillariophyceae were phosphatase negative except *Aulacoseira* sp. (Cao *et al*., 2009).
Dinoflagellates were poor competitors for phosphate accumulation compared to diatoms; they
have to excrete much more APase than diatom to hydrolyze DOP to satisfy their P demand,
even when phosphate is adequate (Rengefors *et al*.,2003). In nutrient addition experiments, a
higher percentage of dinoflagellates were identified with cell-specific APA than diatoms
(Dyhrman *et al*., 2006). It can explain why the $APA_{>3.0\mu m}$ peaked in May when the Pyrrophyta
subdominated the phytoplankton community. It was consistent with the results in Monterey
Bay that dinoflagellates comprised only 14% of all cells counted and accounted for 78% of
APase-producing cells examined (Nicholson *et al*.,2006). As the cell density of Cyanophyta
increased in summer and autumn, the $APA_{>3.0\mu m}$ was also prompted. *Microcystis aeruginosa*
was confirmed can also synthesize APase (Tan *et al*., 2012). The dominating of Cyanophyta
during the summer and autumn resulted in the high amount of APA. The synchronous pattern
of alkaline phosphatase activity and algal cells amount can also be found in Jialing River (Pu
*et al*., 2014). The higher algal cell density in midstream than in estuary can also explain why
the $APA_T$ was higher in midstream. It could be concluded that phytoplankton communities
determined the level of $APA_{>3.0\mu m}$, which determined the significant seasonal and regional
variations of $APA_T$ .
APA showed significant seasonal and regional variations, with lower value in inlet waters
and higher value in the estuarine, and relatively low in winter and high in summer (Jansson *et*





*al*.,1988). However, the distribution characteristics of APA in this study were not consistent
strictly with the above mentioned. The $APA_T$ fluctuated frequently from spring to autumn.
Relative stable level of $APA_T$ in winter can be seen in Figure 2. This phenomenon may result
from the fluctuant water level of the TGR. For the sake of flood control and hydropower, the
water level in the TGR is subjected to the specific management of the TGD and is meant to
seasonally fluctuate between 145 and 175 m a.s.l. It has been demonstrated that the
turbulence promoted the phytoplanktonic APA and accelerated the biogeochemical cycle of P
in Lake Taihu (Zhou *et al*., 2016). This was consistent with our results that the high APA was
present during the significant water level fluctuated period from spring to autumn. Meanwhile,
it has been proved that the APA increased with water temperature (Healey and Hendzel, 1979;
Huber and Kidby, 1984). The positive relationship between WT and APA in this study
(Table.1) supports the conclusion that WT determined the APA through its effects on the
phytoplankton seasonally and the direct influences on APase.

**5. Conclusions**

The size-fractionation of APA indicated that the phytoplankton contributed greatly to APA
production and the spatio-temporal heterogeneity was the characteristics of APA distribution
pattern. The phytoplankton communities with different dominant species and the algal cell
density determined the significant seasonal and regional variations of $APA_T$. Water level and
water temperature were also proved related to the APA's spatio-temporal variation.

*Acknowledgements.* This study has been jointly supported by the National Natural Science
Foundation of China (No: 31123001) and the Science and Technology Research Project of
China Three Gorges Corporation (No. CT-12-08-01). We would like to thank Dianbao Li and
Yi Yang for valuable assistance during the field sampling. The help of Chunxiang Xia, Haiyu
Niu and Wujuan Mi with the laboratory analysis is greatly appreciated.

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
