# Peer review of "Published: 12 December 2016"

_Biogeosciences, 2016_

## Referee Comment (RC1) · Anonymous Referee #1 · 11 Jan 2017

General Comments:

The manuscript has a goal to investigate the importance of phytoplankton to bulk alkaline phosphatase activity (APA) in a tributary of the Three Gorges Reservoir by assessing size fractioned APA and using correlations to infer relation to environmental parameters and phytoplankton communities. There are some fundamental flaws and assumptions made by the authors, which make the relevance of this type of analysis questionable.

First, the authors purport that since APA on size fractions $>3.0\mu$m is greater than that

on 0.45-3$\mu$m, phytoplankton are the main source of APA. There is a wealth of emerging information showing that many (if not all) phytoplankton cells have a host heterotrophic bacteria inhabiting or in close association with cells, making these types of measurements difficult to assign to individual cells alone. Further, many phytoplankton exist in the 1-3$\mu$m size range. At best, the study can show distributions of bulk APA across different size fractions. To assign them to phytoplankton or bacteria requires additional analysis (likely coupled genetic probes and/or ELF). Lastly, to call the <0.45$\mu$m "dissolved" seem suspect as well as many bacteria can slip through a 0.45$\mu$m filter and there are likely significant populations of heterotrophic bacteria inhabiting this size fractioned water.

Secondly, there is a timing issue of when samples were retrieved and when they were analyzed. The methods seem to indicate that samples were collected and then 24 hours later, analyzed. Depending on how the water was stored (which was not indicated in the methods) many of the physiological and biological parameters which were measured (such as chlorophyll, TP, SRP, and COD, and APA) will have dramatically changed in that timeframe. Therefore, what is observed at 24 h post collection will not reflect in situ conditions.

Therefor, any conclusions based upon these methods and assumptions are difficult to interpret.

Specific Comments:

APA method is the same as Wang et al., perhaps the authors should acknowledge that. The sentence structure in many instances needs to be addressed as there are missing words and incorrect use of English language. I have highlighted some instances below.

In the discussion section, it seems the authors suggest that the dominant cyanobacteria was Microcystis. Since this organism exists in colonial form, how were these counted? Further, supporting points above, colonies of Microcystis are inhabited by a host of other organisms including heterotrophic bacteria and in some cases diatoms.

Therefore, when the authors correlate bulk APA to cyanophyta when cyanophyta dominate the community, they inadvertently neglect an important complexity to these communities. Fig 8 – Although I have no way to disprove the authors, based upon this plot it seems suspect that APA>3.0$\mu$m would have a significant positive relationship with cell density

Technical Corrections:

Line 12 – "investigation was" should be "investigations were"

Lines 18 to 19 – "Cyanophta" and "Bacillariophyta" are not "species" but phyla

Line 31 – add a "a" between "hydrolyze" and "broad"

Lines 44 to 46 – I don't believe Nausch says this at all.

Line 81 – How were water samples stored between sample time and analysis 24 hours later?

Line 96 – there seems to be a problem with the PDF here as some of the methods appear outside of the margins

Line 258 – or more likely, increased concentrations of SRP

---

## Referee Comment (RC2) · Anonymous Referee #2 · 12 Jan 2017

Revision on the MS No.: bg-2016-455 entitled "Phytoplankton communities determine the spatio-temporal heterogeneity of alkaline phosphatase activity: evidence from a tributary of the Three Gorges Reservoir" by Yijun Yuan et al. This study investigate the spatial and temporal variation of alkaline phosphatase activity (APA) in different fractions (<0.45 $\mu$ m, 0.45-3  $\mu$ m, >3 $\mu$ m) of waters from the Xiaojiang River (China). This research topic is not novel in the field of aquatic microbial ecology, though eventual interest can rise from the specificities of the studied site. The MS is clear and well written, but problems in the methodology and statistical analyses make me feel rather skeptical on results interpretation by the authors.

Main comments:

The main weak point of this study is methodological. Authors consider bacteria absent in the fraction < 0.45  $\mu m$ , but they can actually be comprised between 0.45-0.2  $\mu m$ . According to this, half of the total APA in your study is found in the dissolved fraction (53.4%, L 235) were a huge amount of bacteria are still present after filtration. I suggest authors to take this comment into account and modify results and discussion sections accordingly.

Hypothesis in this study are missing. Please, supply them at the end of the introduction section.

Statistical analyses: all environmental parameters analyzed in this study were retained, after MonteCarlo's permutation test, in the RDA which is quite surprising. I was really confused after reading P10 L200-202 where authors state that permutation permitted to determine the significance of canonical axes. Could you please clarify this? Second, I suggest authors to remove APA total from the RDA since it will covaries with APA in fractions. Third, is there a sense in discussing about correlations between APA and environmantal parameters showing such a low "r-values"? In my opinion, the correlations presented in Figure 6 and 8 should be removed. Fourth, is there a sense in checking for chl-a and APA<0.45 $\mu$ m correlations when you already know that algae are not present in this fraction? Finally, standard errors over means are not present in Figure 2. Please supply.

Discussion: I suggest authors to include phytoplankton community composition (i. e. diversity indices) in RDA in order to reinforce discussion in P13 L253-277. I would also appreciate a deeper discussion in spatial differences in APA in the Xiaojiang River. Why APA decrease downstream in the estuary zone?

Minor comments:

The amount of replicates analyzed for each of the biological parameters (i. e. APa,

chl-a...) measured has not been specified in the methods section

Be consistent through your MS on: APA or APase? Check APA units (  ${\sf molPNP}$  L-1  ${\sf min-1})$

L27-27. This statement is contradictory according to what you described above (L22-27).

L32-32. Not only cell surfaces but also freely diffusible enzymes (See Burns et al. 2013)

L54-55 and L59-60. Repetition.

L 57-59. A reference is missing.

L68-70. Not clear, rephrase please.

L 96. Correct.

L122-123. Supply reference here, please.

Figure 2. Remove lines indicating seasons, they are confusing. Use dotted line, at least, for Water Level, this will improve lecture.

Show ANOVA P-values in a separate Table for improving clarity of results.

L158. This sentence is not correct since cyanophyta are dominant in April.

L247-248. This statement is wrong. Reconsider it after reading main comments described above.

L270-271. Rephrase, please.

---

## Author Comment (AC1) · 6 Mar 2017

Dear Editors and Reviewer, Thank you very much for your positive and constructive comments on our manuscript. We have carefully made corrections according to the comments, we hope it could meet with approval. Please see the attached point-by-point responses and the tracked change version of manuscript for your further evaluation.

Response to Reviewer's comments: General Comments:

Referee #1: First, the authors purport that since APA on size fractions >3.0$\mu$m is greater than that on 0.45-3$\mu$m, phytoplankton are the main source of APA. There is

a wealth of emerging information showing that many (if not all) phytoplankton cells have a host heterotrophic bacteria inhabiting or in close association with cells, making these types of measurements difficult to assign to individual cells alone. Further, many phytoplankton exist in the 1-3$\mu$m size range. At best, the study can show distributions of bulk APA across different size fractions. To assign them to phytoplankton or bacteria requires additional analysis (likely coupled genetic probes and/or ELF). Lastly, to call the <0.45$\mu$m "dissolved" seem suspect as well as many bacteria can slip through a 0.45$\mu$m filter and there are likely significant populations of heterotrophic bacteria inhabiting this size fractioned water. Response: Yes, we admit the methods of coupled genetic probes and ELF are more accurate than the method of filtration, we will use the two methods to verify our results in future. We know the size fractionation by filtration is never completely absolute (i.e., overlapping size), it is still widely accepted in the field of aquatic ecology because of the limitation of great amounts of water samples and the equipment in situ (Zhou et al.,2016;Wang et al.,2015). The same method on the size fractionation by filtration was used in some previous studies (Cao et al., 2010; Song et al.,2009), this method still provides useful information on the major microorganisms possibly contributing to APA. In this manuscript we used this method according to the previous references and we purport that since APA on size fractions >3.0$\mu$m is greater than that on 0.45-3$\mu$m, phytoplankton are the main source of APA. We mentioned the main source of APA originated from phytoplankton, but phytoplankton not the only one source. We don't ignore the contribution of the host heterotrophic bacteria and overlapping size. For the <0.45$\mu$m "dissolved" one, we also diclare the dissolved is the main type not the only one type. We provided the previous studies in the following: Xiuyun Cao, Chunlei Song, Yiyong Zhou. Limitations of using extracellular alkaline phosphatase activities as a general indicator for describing P deficiency of phytoplankton in Chinese shallow lakes. J Appl Phycol, 2010,22:33–41. Song Chunlei, Cao Xiuyun, Zhou Yiyong. Fluctuation of size-fractionated alkaline phosphatase after bloom disappearance in two shallow ponds. Fresenius environmental bulletin,2009,18(6):982-988. Jian Zhou . Boqiang Qin . Ce′line Casenave . Xiaoxia

Han. Effects of turbulence on alkaline phosphatase activity of phytoplankton and bacterioplankton in Lake Taihu. Hydrobiologia, 2016, 765:197–207. Peifang Wand, Lingxiao Ren,Chao Wand, Jin Qian, Jun Hou. Presence and patterns of alkaline phosphatase activity and phosphorus cycling in natural riparian zones under changing nutrient conditions. J. Limnol., 2015,74(1): 155-168.

Referee #1: Secondly, there is a timing issue of when samples were retrieved and when they were analyzed. The methods seem to indicate that samples were collected and then 24 hours later, analyzed. Depending on how the water was stored (which was not indicated in the methods) many of the physiological and biological parameters which were measured (such as chlorophyll, TP, SRP, and COD, and APA) will have dramatically changed in that timeframe. Therefore, what is observed at 24 h post collection will not reflect in situ conditions. Therefore, any conclusions based upon these methods and assumptions are difficult to interpret. Response: Thanks for your comment and sorry for our unclear expression. We have rewritten this section and some information was provided in detail. The water samples for APA test were filtered immediately after collection in situ, the filters were put into a portable refrigerator at 0 oC and analyzed within 24 h. In order to avoid the physiological and biological parameters changed dramatically, all water samples for the other parameters measurement were also stored in a portable refrigerator at 0 oC after collected. Therefore, the parameters can reflect in situ conditions. So the conclusions based upon these methods are reliable (See L92-96).

Specific Comments: Referee #1: APA method is the same as Wang et al., perhaps the authors should acknowledge that. Response: Sorry for our carelessness. We have added the part in the Acknowledgements (please see line L337).

Referee #1: In the discussion section, it seems the authors suggest that the dominant cyanobacteria was Microcystis. Since this organism exists in colonial form, how were these counted? Further, supporting points above, colonies of Microcystis are inhabited by a host of other organisms including heterotrophic bacteria and in some

cases diatoms. Therefore, when the authors correlate bulk APA to cyanophyta when cyanophyta dominate the community, they inadvertently neglect an important complexity to these communities. Response: Yes, you are right. One difficulty we encountered in the phytoplankton counts was caused by the Microcystis colonies. In order to calculate the biomass of phytoplankton as accurately as possible, a rapid, high-speed blending method for disrupting colonies of Microcystis aeruginosa to single cells in preparation for cell counts was employed. This sample preparation method was proved rapid and convenient for counting M. aermginosa and associated organisms(Tamar and Arcangela,1987). Therefore, the result of the correlation analysis based on the bulk APA to cyanophyta was reliable. On the other hand, Classically, colonies of Microcystis are inhabited by a host of other organisms including heterotrophic bacteria and in some cases diatoms. In this study, you can find that when Microcystis colonies dominated the community, the relative abundance of diatom is very low (Fig.4). As the heterotrophic bacteria, water samples were filtered after the strong oscillation, many heterotrophic bacteria was peeled by the shearing force. Due to the heterotrophic bacteria and diatom is not the major contributor, we think the bulk APA to cyanophyta when cyanophyta dominate the community. The related references: Tamar Zohary, Arcangela M. Pais Madeira. Counting natural populations of microcystis aeruginosa: a simple method for colony disruption into single cells and its effect on cell counts of other species. J. Limnol. SOC. sth. Afr. 1987,13(2):75-77.

Referee #1: Fig 8 – Although I have no way to disprove the authors, based upon this plot it seems suspect that APA>3.0$\mu$m would have a significant positive relationship with cell density Response: Yes, what you said is correct. We check the data again and find there is no significant relationship between APA>3.0$\mu$m and cell density. This result indicated the species-specific of the phytoplankton excreting alkaline phosphatase.

Technical Corrections: Referee #1: Line 12 – "investigation was" should be "investigations were" Response: Agreed and revised (please see L13).

Referee #1: Lines 18 to 19 – "Cyanophta" and "Bacillariophyta" are not "species" but

phyla Response: Agreed and revised (please see L19).

Referee #1: Line 31 – add a "a" between "hydrolyze" and "broad" Response: Agreed and revised (please see L32).

Referee #1: Lines 44 to 46 – I don't believe Nausch says this at all. Response: Agreed and revised (please see L44-47).

Referee #1: Line 81 – How were water samples stored between sample time and analysis 24 hours later? Response: In order to avoid the physiological and biological parameters changed dramatically, the samples were stored in portable refrigerator after collected, all samples were analyzed within 24 h.

Referee #1: Line 96 – there seems to be a problem with the PDF here as some of the methods appear outside of the margins Response: Agreed and revised (please see L107).

Referee #1: Line 258 – or more likely, increased concentrations of SRP Response: Thanks for your comment. According to your suggestion, we have rewritten this sentence. The increase concentration of SRP isn't contradictory to the decrease of algal cell density of phytoplankton. SRP is the bioavailable form of phosphorus that the phytoplankton can uptake directly. As the cell density decreased in winter, the concentrations of SRP increased. So the low APA>3.0$\mu$m in winter may result from the low algal cell density of phytoplankton and the increased concentrations of SRP in parallel.

Please also note the supplement to this comment:
http://www.biogeosciences-discuss.net/bg-2016-455/bg-2016-455-AC1-supplement.pdf

---

## Author Comment (AC2) · 6 Mar 2017

Dear Editors and Reviewer, Thank you very much for your positive and constructive comments on our manuscript. We have carefully made corrections according to the comments, we hope it could meet with approval. Please see the attached point-by-point responses and the tracked change version of manuscript for your further evaluation.

Response to Reviewer's comments: Main comments: Referee #2: The main weak point of this study is methodological. Authors consider bacteria absent in the fraction $< 0.45\mu$m, but they can actually be comprised between 0.45-0.2$\mu$m. According to this,

half of the total APA in your study is found in the dissolved fraction (53.4%, L235) were a huge amount of bacteria are still present after filtration. I suggest authors to take this comment into account and modify results and discussion sections accordingly. Response: Yes, we agree with you that the fraction $< 0.45\mu$m contains some pico-bacteria between 0.45-0.2$\mu$m and some picophytoplankton. The fraction $< 0.45\mu$m can't be called as the dissolved fraction though many articles named it as the dissolved fraction. We changed it as the picoplankton/dissolved fraction.

Referee #2: Hypothesis in this study are missing. Please, supply them at the end of the introduction section. Response: Agreed and revised (please see L70-72).

Referee #2: All environmental parameters analyzed in this study were retained, after MonteCarlo's permutation test, in the RDA which is quite surprising. I was really confused after reading P10 L200-202 where authors state that permutation permitted to determine the significance of canonical axes. Could you please clarify this? Response: Thanks for your comment and sorry for our unclear expression. The environmental parameters listed in Table 1 were proved correlated with APA fractions. But the parameters such as total nitrogen (TN), nitrate nitrogen (NO3-N) and ammonium nitrogen (NH4-N) were proved not correlated with APA, so these parameters were excluded after MonteCarlo's permutation test. That is why all environmental parameters analyzed in Table 1 were retained, after MonteCarlo's permutation test. The RDA analysis in our study was done in strict accordance with the steps described in the book "Multivariate analysis of ecological data using CANOCO". It was illustrated that the permutation tests can be used to test virtually any relationship. To illustrate its logic, the permutation tests are used for testing the significance of a regression model. References:Lepš J, Šmilauer P. Multivariate analysis of ecological data using CANOCO[M]. Cambridge university press, 2003.

Referee #2: Second, I suggest authors to remove APA total from the RDA since it will covaries with APA in fractions. Response: Agreed and revised (please see L229).

Referee #2: Third, is there a sense in discussing about correlations between APA and environmental parameters showing such a low "r-values"? In my opinion, the correlations presented in Figure 6 and 8 should be removed. Response: The source of alkaline phosphatase is complicated. It may excrete from phytoplankton, bacteria, zooplankton, sediment and so on. As to the phytoplankton, the species-specific result in the related environmental parameters to the alkaline phosphatase differed. Besides, the previous studies proved the correlation relationship between APA and SRP emerged within a certain threshold. These uncertainties lead to the correlations between APA and environmental parameters showing such a low "r-values". Although the "r-values" was low, the results of correlated analysis and RDA analysis were consistent. So both the correlations presented in Figure 6 and 8 and the RDA analysis in Figure 7 can prove the interaction between APA and the environmental parameters.

Referee #2: Fourth, is there a sense in checking for chl-a and APA<0.45$\mu$m correlations when you already know that algae are not present in this fraction? Response: Yes, you are right. This part was deleted.

Referee #2: Finally, standard errors over means are not present in Figure 2. Please supply. Response: Agreed and revised (please see L152).

Referee #2: I suggest authors to include phytoplankton community composition (i. e. diversity indices) in RDA in order to reinforce discussion in P13 L253-277. I would also appreciate a deeper discussion in spatial differences in APA in the Xiaojiang River. Why APA decrease downstream in the estuary zone? Response: Thanks for your good suggestion. The species-specific of phytoplankton excreting alkaline phosphatase was meaningful and interesting. We conducted the research that focus on the relationship between phytoplankton community composition and alkaline phosphatase, and found some interesting results (unpublished). This study focuses on the spatio-temporal heterogeneity of alkaline phosphatase activity. So it is difficult for us to explain the meaningful topic mentioned above explicitly. As to the APA decrease downstream in the estuary zone, it results from the higher algal cell density in midstream than in estuary

that has been illustrated in P13 (L307-310).

Minor comments: Referee #2: The amount of replicates analyzed for each of the biological parameters (i. e. APa, chl-a...) measured has not been specified in the methods section. Response: Agreed and revised (please see L92).

Referee #2: Be consistent through your MS on: APA or APase? Check APA units ( molPNP L-1min-1) Response: Thanks for your comments. APA is different from APase. APA means alkaline phosphatase activity (L13), and APase means alkaline phosphatase (L32).

Referee #2: L27-27. This statement is contradictory according to what you described above (L22-27). Response: This study came to a conclusion that phytoplankton communities determine the spatio-temporal heterogeneity of alkaline phosphatase activity (L23-24). Besides this, water temperature was proved to be positively correlated with APA and water level (WL) were negatively correlated with APA (L27-28). So it was concluded that spatio-temporal heterogeneity of APA was also related to water temperature and hydrodynamics.

Referee #2: L32-32. Not only cell surfaces but also freely diffusible enzymes (See Burns et al. 2013) Response: Agreed and revised (please see L34).

Referee #2: L54-55 and L59-60. Repetition. Response: Agreed and revised

Referee #2:L 57-59. A reference is missing. Response: Agreed and revised (please see L61).

Referee #2:L68-70. Not clear, rephrase please. Response: Agreed and revised (please see L70-72).

Referee #2: L 96. Correct. Response: Agreed and revised (please see L108).

Referee #2:L122-123. Supply reference here, please. Response: Agreed and revised (please see L134).

Referee #2: Figure 2. Remove lines indicating seasons, they are confusing. Use dotted line, at least, for Water Level, this will improve lecture. Response: Agreed and revised (please see L152).

Referee #2: Show ANOVA P-values in a separate Table for improving clarity of results. Response: All the P-values listed in this study were less than 0.05, some of them were even less than 0.01. So the correlation analysis results were reliable. It is sufficient to prove the relationship between APA and the environment parameters.

Referee #2:L158. This sentence is not correct since cyanophyta are dominant in April. Response: Agreed and revised (please see L174).

Referee #2:L247-248. This statement is wrong. Reconsider it after reading main comments described above. Response: Agreed and revised (please see L247-253).

Referee #2:L270-271. Rephrase, please. Response: Agreed and revised (please see L275-279).

Please also note the supplement to this comment:
http://www.biogeosciences-discuss.net/bg-2016-455/bg-2016-455-AC2-supplement.pdf

———————————————

---

## Author Comment (AC4) · 6 Mar 2017

[revised manuscript text omitted]

Water temperature (WT), pH, dissolved oxygen (DO) and conductivity (Cond.) were 86 measured using a YSI model Professional Plus multiparameter probe (USA); Transparency 87 (SD) was measured with a Secchi disk; and turbidity (Turb.) was measured with a WGZ-B 88 turbidmeter (XinRui, Shanghai). Water level (WL) was recorded by GPS in situ. Surface 89 90 water samples (0.5m) were collected with a Van Dorn sampler at seven sampling sites (XJ, 91 HS02, HS01, GY02, GY01, QM02, QM01) (Fig.1) monthly from October 2013 to September 92 2014. All samples were run in triplicate. In order to avoid the physiological and biological 93 parameters changed dramatically, The water samples for APA test were filtered immediately after collection and strong oscillation in situ, the filters were put into a portable refrigerator at 0 °C 94 95 and analyzed within 24 h. All water samples for the other parameters measurement were also 96 stored in a portable refrigerator at 0 °C after collected and tested within 24 h. Concentrations of chlorophyll a (Chl a), total phosphorus (TP), soluble reactive phosphorus (SRP), chemical 97 98 oxygen demand (COD) were analyzed after samples collected within 24h. Samples for quantitative phytoplankton analyses were fixed with neutral Lugol's solution, and 99 100 concentrated after 48 h sedimentation (Utermohl, 1931).

Figure1. Maps of the location of the Three Gorges Reservoir Region, and the sampling sites in the Xiaojiang River

105 2.2 Measurement of APA

101

102 103

104

106 APA was measured using a modified procedure (Gage and Gorham, 1985; Boon, 1989). A total of 2ml water samples were incubated at 37°C for 4h in the presence of 1ml 0.05M 107 Tris-HCl buffer (pH=8.5) and 2ml 0.3mM p-nitronphenylphosphate (p-NPP) as substrate, 108 subsequently, 0.1ml 0.1M NaOH was added into the mixture after 4h. The release of 109 p-nitrophenol from p-nitrophenylphosphate was determined by absorbance at 410nm using a 110 spectrophotometer (TU-1810), and APA was calculated in nM·L-1·min-1. APA was 111 determined in unfiltered water (APAT) and water samples filtered through 0.45 (the 112 picoplankton/dissolved alkaline phosphatase activity, APA<0.45um) and 3.0µm membrane 113 filters (APA<3.0µm). The activity in algal fraction (APA>3.0µm) and in bacterial fraction 114 115 (APA0.45-3.0µm) were calculated as follows: APA>3.0µm= APAT - APA<3.0µm, APA0.45-3.0µm= APA<3.0µm—APA<0.45µm (Chrost *et al.*, 1984). 116

**117 2.3 Measurement of SRP, Chl *a* , TP, COD and phytoplankton quantification**

Water samples used for the Chl a measurement were filtered with Whatman GF/C filter, 118 then the residuals on the filter were extracted using 90% acetone solution in the darkroom for 119 24 h at 4°C, and Chl a was analyzed spectrophotometrically (A.P.H.A, 1995). The 120 121 concentrations of SRP were measured after all water samples were filtered through pre-washed filters (Whatman GF/C, glass microfiber filters). The concentrations of SRP, total 122 123 phosphorus (TP) and chemical oxygen demand (COD) were analyzed according to the 124 standard methods (A.P.H.A, 1995). Phytoplankton was quantified at 400× magnification with a light microscope (OLYMPUS BX41). The identification of phytoplankton species is 125

according to Hu and Wei (Hu and Wei, 2006).

**127 **2.4 Statistical analysis**

Statistical analysis was carried out using the SPSS 13.0 package. Variance analysis 128 (one-way ANOVA) was used to compare the means of APA in different seasons and 129 sampling sites. Non-parametric correlation (Spearman) analyses were employed for 130 determining relationships among APA<0.45um, APA0.45-3um, APA>3.0um, APAT and the 131 132 environmental factors. Detrended correspondence analysis (DCA) of the size-fractionated APA and environmental data was performed using CANOCO version 4.5 to determine 133 134 whether linear or unimodal ordination methods should be applied (Zhu et al, 2013). Before 135 the analysis, the abiotic and biological data were transformed by log(x+1). Redundancy 136 analysis (RDA) was performed to get an approximate ordering of the size-fractionated APA's 137 optima for environmental variables. The significance of canonical axes and environmental variables to explain the variance of the size-fractionated APA was tested using Monte Carlo 138 139 simulations with 499 permutations.

140 **3. Results**

**141 **3.1 APAT distribution pattern**

The APAT ranged from 1.19-47.6 nmol·L-1·min-1(Fig.2). The lowest level of APAT was observed in winter. Besides, the average APAT in summer and autumn were significantly higher than in other seasons (P<0.05). Meanwhile, significant difference between summer and autumn were not detected (P>0.05). The mean water level was high in winter(169.7±4.5 m) and low in summer(149.3±3.1 m), the variations of water level presented different trends with that of APAT at temporal scales.

The highest value of annual average  $APA_T$  in GY01 and lowest in XJ were also showed in Fig.2. No difference of  $APA_T$  among the seven sites was observed in winter and spring (*P*>0.05). The average  $APA_T$  of GY01 in summer and autumn are significantly higher than those of HS02 and XJ (*P*

---

## Author Comment (AC5) · 6 Mar 2017

The comment was uploaded in the form of a supplement:
http://www.biogeosciences-discuss.net/bg-2016-455/bg-2016-455-AC5-supplement.pdf